# Food anaphylaxis diagnostic marker compilation in machine learning design and validation

**Inderpal S. Randhawa** [ID]*, **Kirill Groshenkov**[◎†], **Grigori Sigalov**[◎]

Food Allergy Institute, Long Beach, California, United States of America

◎ These authors contributed equally to this work.
† Deceased.
* docrandhawa@tpirc.org

## Abstract

**Data Availability Statement:** The complete codes and data necessary to reproduce our calculations and verify our findings are available on GitHub (https://github.com/TPIRC/ai_paper_2022).

### Background

Traditional food allergy assessment of anaphylaxis remains limited in accuracy and accessibility. Current methods of anaphylaxis risk assessment are costly with low predictive accuracy. The Tolerance Induction Program (TIP) for anaphylactic patients undergoing TIP immunotherapy produced large-scale diagnostic data across biosimilar proteins, which was used to develop a machine learning model for patient-specific and allergen-specific anaphylaxis assessment. In explanation of construct, this work describes the algorithm design for assignment of peanut allergen score as a quantitative measure of anaphylaxis risk. Secondarily, it confirms the accuracy of the machine learning model for a specific cohort of food anaphylactic children.

### Methods and results

Machine learning model design for allergen score prediction utilized 241 individual allergy assays per patient. Accumulation of data across total IgE subdivision served as the basis of data organization. Two regression based Generalized Linear Models (GLM) were utilized to position allergy assessment on a linear scale. The initial model was further tested with sequential patient data over time. A Bayesian method was then used to improve outcomes by calculating the adaptive weights for the results of the two GLMs of peanut allergy score prediction. A linear combination of both provided the final hybrid machine learning prediction algorithm. Specific analysis of peanut anaphylaxis within one endotype model is estimated to predict the severity of possible anaphylactic reaction to peanut with a recall of 95.2% on a dataset of 530 juvenile patients with various food allergies, including but not limited to peanut allergy. Receiver Operating Characteristic analysis yielded over 99% AUC (area under curve) results within peanut allergy prediction.

**Funding:** The authors received no specific funding for this work.

**Competing interests:** The authors have declared no competing interests exist.

## Conclusions

Machine learning algorithm design established from comprehensive molecular allergy data produces high accuracy and recall in anaphylaxis risk assessment. Subsequent design of additional food protein anaphylaxis algorithms is needed to improve the precision and efficiency of clinical food allergy assessment and immunotherapy treatment.

## Introduction

Food allergy and anaphylaxis remain a high prevalence disorder in developed countries [1]. With an estimated 6% of children diagnosed with food anaphylaxis, the diagnosis burden encompasses both patients and families [2]. The cost of living with food anaphylaxis is $24.8 (95% CI, $20.6-$29.4) billion annually ($4184 per year per child) [3]. Of greater concern, the incidence of food allergy is increasing in both developed and developing countries [4].

Proper diagnosis of food anaphylaxis has evaded singular test methods. The classical method of skin prick testing at best maintains 50–60% positive predictive value [5]. Serum Immunoglobulin E (sIgE) testing via older RAST (radioallergosorbent test) methods or more recent Immunocap technology maintains over 50% false positivity in food allergy diagnosis [6]. In the last two decades, component resolved diagnostic testing held promise in greater anaphylaxis predictive accuracy. However, large studies reveal limitations in any improvement of accuracy of sIgE [7]. The gold standard of diagnosis remains the double-blind placebo controlled food challenge (DBPCFC). Interestingly, even patients who pass a DBPCFC are able redevelop anaphylaxis to that "passed" food challenge protein [8].

Our institute has developed a novel approach to the predictive diagnosis of food anaphylaxis followed by induction of sustained unresponsiveness of primary food allergens. The Tolerance Induction Program (TIP) involves utilization of large-scale data analytics assessing hundreds of pro allergic markers (skin prick testing, component resolved diagnostics, immunocap specific IgE, histamine release assay, peripheral eosinophil count) as well as tolerance markers (cytokine profiles, IgG4 specifics, total IgG4, total IgG) across collected patient cohorts since 2007. The machine learning process organizes an incoming patient's specific markers against the molecular groupings of the database. Once analyzed, the patient's markers are assessed into an endotype and a subsequent risk assessment of food allergens and biosimilar proteins is organized. This organized "snapshot" of an individual patient is further utilized to build food protein dose exposure cycles based on standardized eliciting doses of food proteins. Repeat marker analysis is monitored to assess the final process of tolerance induction and one-week sustained unresponsiveness for all primary allergens. The TIP model has been previously published [9].

The goal of this study is to overcome the limitations of current food allergy predictive models. The endpoint of TIP is high dose food allergen protein exposure on a minimum of a weekly basis with continued reduction in biosimilar food allergy markers. The severity of allergic reaction is quantitatively characterized by an allergen score. The databases and extracted formats to achieve these results served as the basis for machine learning development. A machine learning algorithm was developed to predict the allergen score using lab test data as features. The described algorithm was built and optimized to achieve > 95% recall. The training and testing data model are further described specific to clinical outcomes defined as sustained unresponsiveness.

## Materials and methods

### Data

Data sets are organized into two primary groups. The first group involves all molecular data which was consistently acquired from each patient enrolled in the program since 2007. The data set includes extended profiles of component resolved diagnostics, skin prick testing, and sIgE to all primary plant and animal protein regardless of the patient's clinical history. The second data set includes immune system markers consistently acquired from each patient enrolled in the program since 2007. Markers include B cell antibody diversity, cytokine production, basophil indices, eosinophil counts, and lymphocyte counts. The final data set includes the similarity of plant and animal species allergens utilizing data adopted from public domains such as the food allergen datasets Allergen Nomenclature (http://www.allergen.org/index.php, accessed on 9 October 2020) and the Structural Database of Allergenic Proteins (SDAP) (http://fermi.utmb.edu/SDAP/). All datasets have been rigorously screened, and there is no duplication between positive and negative samples.

### A machine learning algorithm for allergen score prediction

To provide a physician with a tool for quick and accurate evaluation of the patient's risk of anaphylaxis and the severity of a possible allergic reaction to peanut, we developed a Machine Learning (ML) algorithm. The algorithm is using a set of lab test data with up to 241 individual tests, as well as the patient's personal information such as age and weight, as feature candidates. In the process of model optimization and training, relevant features are selected via iterations of a greedy selection algorithm, while features not meeting the standard relevance criteria are dropped. This automatic, data-centric process eliminates a human factor in deciding which test results should or should not be used in the ML model design. The larger set of tests for possible allergic proteins is available, the higher accuracy of the ML algorithm could theoretically be achieved. The relevant features are assigned adoptive weights as the result of model training. The training data set is tabular, based on depersonalized patient data, and will be described in detail below. Once trained, the ML model is used for new (out of set) patients to predict a quantified expected patient's allergic reaction to peanut. The physician may then consider the predicted value while finalizing the diagnosis and selecting the treatment.

The severity of allergic reaction to peanut in a particular patient is normally classified as Anaphylactic (Ana), Sensitized (Sen), or Tolerant (Tol). Additionally, within each of these three classes, it is characterized by a positive number up to 100 referred to as an allergen score. The larger the number, the stronger the allergic reaction and, consequently, the risk of anaphylaxis. The score is usually rounded so that there could be up to 5 discrete severity levels within each severity class, for example Ana20, Ana40, Ana60, Ana80, Ana100. Since, for example, Sen100 is the highest severity level in the class Sen, and Ana20 is the lowest severity level in the class Ana that is directly above class Sen, it is natural to use a continuous absolute linear scale from 0 to 300 for the allergen score. Scores up to 100 correspond to identical scores within class Tol; scores between 120 to 200 correspond to Sen20 . . . Sen100; and finally, scores between 220 to 300 correspond to Ana20 . . . Ana100. These score representations are equivalent, each having its own advantages. In what follows, we will use both notations interchangeably.

To be clear, the ML algorithm is used as an auxiliary recommendation tool to assist a qualified human medical professional, who may consider the algorithm's prediction in a clinical decision-making process ultimately left to the physician. Given the severity of anaphylaxis, and the fact that it is safer to overestimate rather than underestimate the risk of anaphylaxis,

the algorithm is optimized, through its objective function's design, to have the highest accuracy for class Ana and the upper levels of class Sen, and to err toward higher allergen scores rather than uniformly. In technical terms, positive errors (risk overestimation) have smaller weights in weighted Root Mean Square Error (RMSE) summation than negative errors do.

## Dataset and features

For the purposes of this paper, we only focus on peanut allergy for endotype 1 patients. However, the same approach is applicable to other proteins and endotypes (results to be published elsewhere). The selected dataset included 530 patients, out of which 340 had total IgE < 1000 kU/L and 190 had total IgE > 1000 kU/L, thus representing two cohorts with different levels of susceptibility to food allergies. Fig 1 shows the distribution of total IgE values for all patients whose data were used in this study.

The available dataset consists of depersonalized patient profiles along with the results of various laboratory tests performed before and during treatment. Those data provide the (potential) input features (independent variables, also called predictors) for a predictive ML algorithm. For each patient, three types of data are available:

- patient's personal information, including age and weight;

- the results of up to 241 blood and skin lab tests performed;

- peanut allergy severity classes and scores (feature "Peanut" on the source data table) as determined by a qualified medical professional (these are the dependent variables whose ground truth values are used to train the ML model).

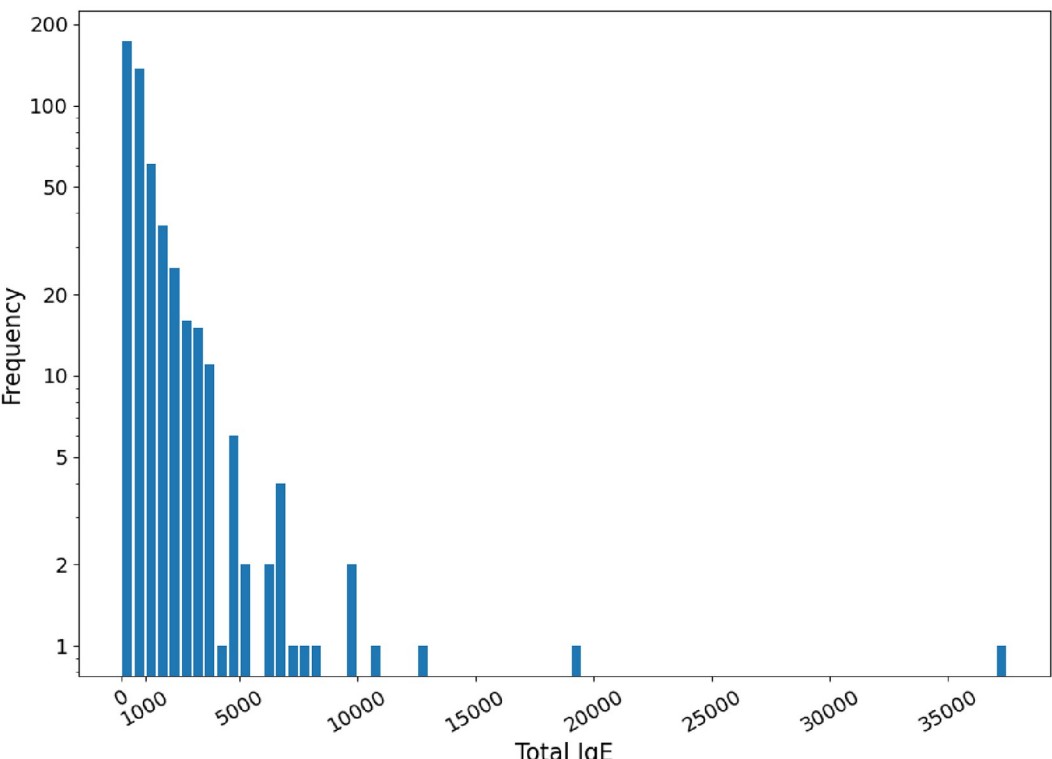

**Fig 1. Distribution of total IgE values over the selected patient dataset.**

The lab test results are usually represented by a nonnegative number within a certain range, with some notable exceptions. First of all, certain features cannot be measured reliably near the lower and/or upper limits of their range; such results may be shown as "less than" or "greater than" a certain threshold value, with a numerical value in the intermediate range where an accurate measurement is possible. Secondly, some of the tests have Boolean or integer-valued underlying nature and could be better treated as binary, categorical, or ordinal variables, rather than as continuous variables. Lastly, the values for many tests are missing, usually because those tests were deemed (by a physician) redundant or unnecessary, considering the patient's anamnesis. Thus, only the ML methods that are capable of dealing efficiently with massively missing data should be considered for the algorithm. Fig 2 illustrates the range and distribution of values for some representative features.

## Algorithm design

As noted above, there are two ways to characterize the allergen score: either as a class (Tol, Sen, Ana) and further as a level 20, 40, . . ., 100 within that class (e.g., Ana40), or as a continuous variable from 0 to 300, rounded to the nearest 20. The step of 20 is chosen for convenience only and does not affect the algorithm as such. Accordingly, there are two possibilities for the design of a predictive ML algorithm. To reflect the former representation, one could build a cascade of a 3-class classifier and a regression model for the level within the class. For the latter representation, a regression model for the continuous score that spans all three classes could be constructed.

For the former choice, to assign one of the 3 classes, one could take into account the fact that those classes are actually ordered (Tol < Sen < Ana in terms of the severity of the allergic reaction), and thus use ordinal regression. Alternatively, one could ignore that fact and use pure classification into categorical variables, expecting that the ML model would learn on its own to assign classes in the correct order. In what follows, we will generate ML models based on each of those design solutions and compare their outcomes.

We began by creating a regression-based Generalized Linear Model (GLM) with link functions to mimic the nonlinear behavior or a hypothesized type of distribution (normal or Poisson distribution, as indicated by the data analysis) for some of the features. We chose a continuous score ranging from 0 to 300 as the dependent variable and rounded the obtained real-numbered predictions to the nearest 20.

Based on biomedical information, certain features are known a priori to be possibly relevant for peanut allergy prediction. The GLM regression model is initially trained using those a priori features, while estimating their actual relevance. After an automatic feature elimination process via iterations of a greedy algorithm, the relevant features are found to be peanut_ige, peanut_igg4, peanut_spt, arah_1, arah_2, arah_3, arah_8, arah_9. While higher numerical values of these features usually imply a higher peanut allergy score, the dependence of the latter on the features is non-trivial and extremely noisy, as illustrated by Fig 3, and requires an advanced ML model to allow for meaningful prediction.

The model generated using the abridged (known a priori) set of features might be improved by considering additional features provided by lab tests for proteins not obviously biosimilar to peanut, and thus not deemed a priori relevant. The model is therefore trained further, trying one additional candidate feature at a time (out of 243 available features, including patient's age and weight). A feature that improves the model is preserved, and then another iteration over candidate features is performed, until no further improvement can be achieved. Some examples of features that are not directly related to peanut but might still improve the ML model that predicts the peanut allergy score are total_igg4, cyt, and eos (Eosinophil count). Table 1

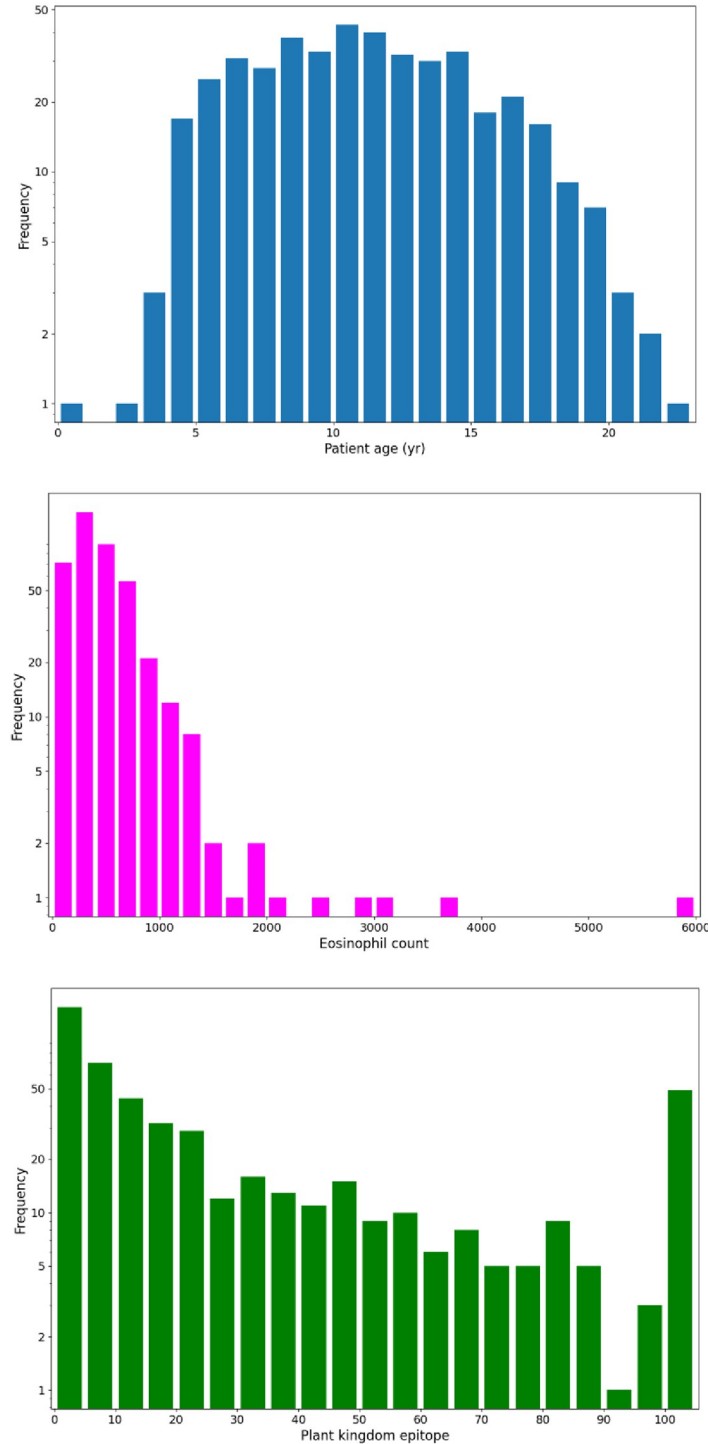

**Fig 2. Distribution of values for some independent data features.**

shows how addition of these non-obvious features, selected automatically through greedy incremental feature testing and addition or rejection, improves the model. The prediction error is estimated using the Leave-One-Out (LOO) cross-validation method. Table 2 shows the top 10 features of the expanded set along with their normalized importance values. I can be

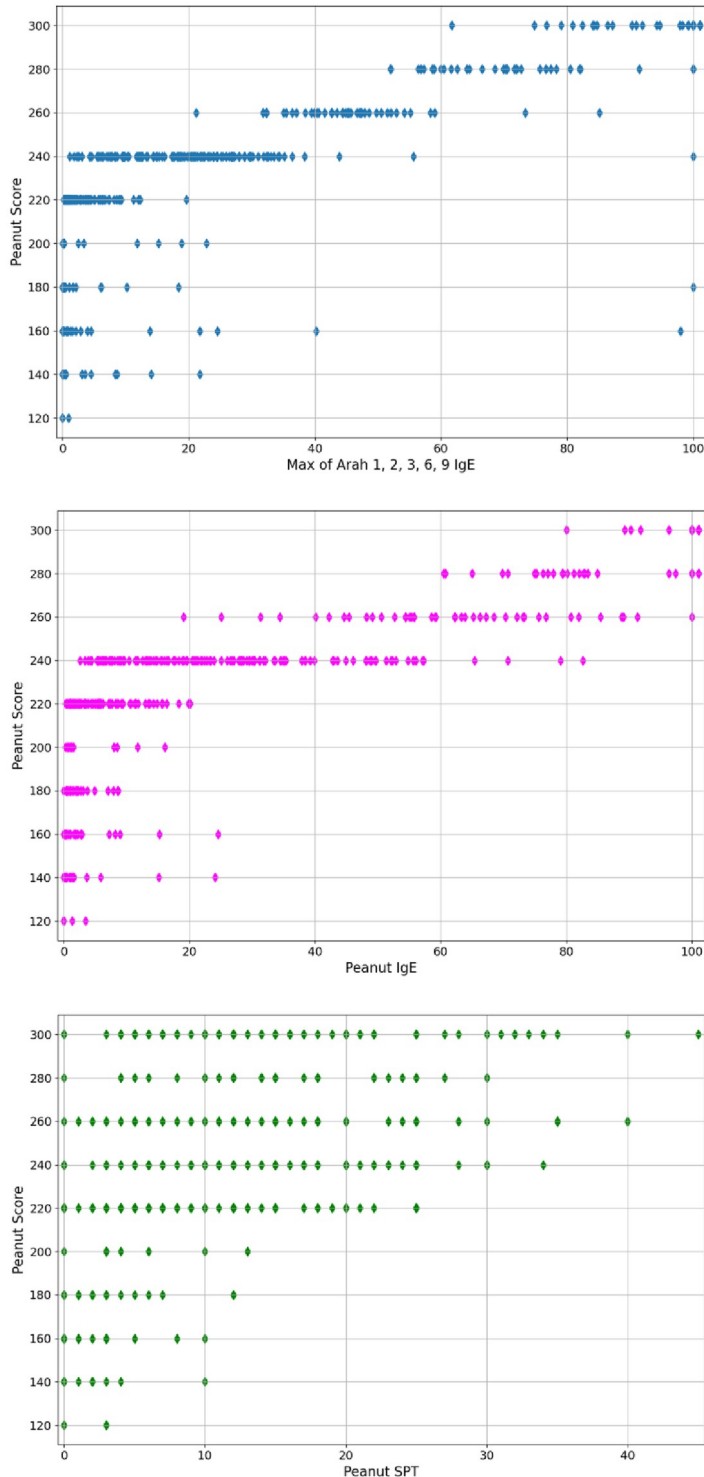

**Fig 3. Correlation between the dependent variable (peanut allergy score) and some relevant predictors.** Absolute peanut allergy scores 120–200 can be interpreted as Sen20 to Sen100, and scores 220–300 as Ana20 to Ana100. By design, arahs = max(arah_1, ..., arah_9).

**Table 1. Mean absolute error (MAE) of the peanut allergy score prediction in ML models with an abridged (a priori known) set of features and an expanded set of features, each optimized to include only relevant features.** Ground truth values are multiples of 20. LOO cross-validation is used.

| Features | class Ana | class Sen |
|---|---|---|
| A priori set | 8 | 25.8 |
| Expanded set | 7.6 | 21.7 |

**Table 2. Top 10 independent features of the expanded set of predictors.**

| Feature | Importance |
|---|---|
| arah_2 | 0.531862 |
| peanut_ige | 0.379455 |
| arah_1 | 0.025432 |
| arah_3 | 0.012586 |
| arah_8 | 0.010985 |
| arah_9 | 0.009214 |
| coconut_ige * | 0.008989 |
| coconut_spt * | 0.00748 |
| peanut_igg4 | 0.007411 |
| peanut_spt | 0.006586 |

* Indicates features not present on the a priori set. Importance values are normalized and meant to indicate a relative importance of the features rather than the weights of the variables.

seen that this list includes all 8 a priori features, but some of the a priori features are less important than some of new features that are not directly linked to peanut.

The mean absolute error (MAE) of the above models is well under 20, which is the distance between the adjacent values of the allergy score's ground truth values. This means that the ML algorithm predicts the allergy severity level to within one step of the ladder, on average. However, it might not always predict the correct class (Ana, Sen, or Tol) when the ground truth level is at or near the boundary between classes (100 or 200). For example, it might predict level Sen100 when the ground truth is Ana20, or vice versa. While the numerical difference between these scores is still only 20, the medical implications of the misclassification (Sen vs Ana) may be serious. We addressed this issue by predicting the class first, and the level within each class second. We compared two approaches for that classification task, one through pure classification and the other through ordinal regression, in which the information about class ordering is used extensively. In both models, we used tree-based methods trained with gradient boosting. The ordinal regression method turned out to be more accurate and was selected for this task. The performance of this method is shown in Table 3. Note that there were no

**Table 3. Classification model performance in ML models with an abridged (known a priori) and expanded sets of features, each optimized to include only relevant features.** Both models employed ordinal regression and were trained with Gradient Boosting for peanut allergy classification into reaction severity classes. Class Tol is not present in the input dataset.

| Features | class Ana | | class Sen | |
|---|---|---|---|---|
| | Precision, % | Recall, % | Precision, % | Recall, % |
| A priori set | 82.3 (303/368) | 98.0 (303/309) | 91.2 (125/137) | 69.0 (125/181) |
| Expanded set | 83.0 (304/366) | 98.3 (304/309) | 91.4 (127/139) | 70.2 (127/181) |

patients tolerant to peanut on the chosen dataset, therefore only classes Ana and Sen are present in the results. However, the ordinal regression method we used is not limited to 2-class classification as it is not a binary or logistic classification by its nature. Classification into 3 (or more, for that matter) classes may be effectively performed using the same method for other food proteins or groups of patients.

Finally, we trained the GLM regression model described above on the data for each class separately, thus achieving both a more accurate classification and more accurate prediction of the level with the class. However, the original full-range (0 to 300) GLM regression model still could be used to improve the final outcomes, because it is trained on a larger dataset. We combined both GLMs by using a Bayesian method to calculate the adaptive weights for the results of the two methods of peanut allergy score prediction. A linear combination of them provided the final hybrid ML prediction algorithm. Its design is summarized in Fig 4.

The accuracy of the final ML algorithm was characterized with Receiver Operating Characteristic (ROC) curves. To verify that the algorithm performs well for both highly susceptible and less susceptible patients, we calculated the ROC curves for 340 patients with total IgE < 1000 and 190 patients with total IgE > 1000. The results shown in Fig 5 demonstrate that the proposed ML algorithm could be used as a diagnostic support tool for a physician to evaluate the expected severity of patient's allergic response to peanut.

## Discussion

Machine learning algorithm design in modern medicine largely endorses imaging aspects of medical specialties such as radiology and ophthalmology. ML accuracy and recall in imaging associated medical specialties continue to improve as reflected in the regulatory approval of such devices. The specialty of Allergy and Immunology to date has no diagnostic machine learning algorithm published in any of the atopic disease states: eczema, asthma, rhinitis or food anaphylaxis. The Tolerance Induction Program has structured its clinical therapeutic model upon data collection across immunological, phylogenetic and clinical markers. The totality of these markers are organized to study food anaphylaxis patients into specific endotype cohorts. Once identified, the cohort specific analysis utilizes machine algorithms based on aforementioned data sets to determine food protein specific anaphylaxis risk. Furthermore, the risk assessment produced serves as the basis for additional machine learning systems producing food protein-based immunotherapy schedules on a patient specific basis. Ultimately, the risk assessment is re-calculated per annum as the patient progresses through large challenges of food proteins in challenge sequences. Upon completion, the patient consumes between 10–30 grams of food specific protein in one sitting to all of once anaphylactic foods. In subsequent weeks, the consumption of this large amount of food protein occurs on a weekly or biweekly basis thus achieving sustained unresponsiveness while freely consuming once anaphylactic foods.

Our machine learning algorithm is designed utilizing anaphylaxis clinical, diagnostic and eliciting dose data. Such data in an anaphylaxis cohort is critical as it reflects known risk of the disease in a large cohort. We defined a risk score as a continuous variable, which effectively becomes ordinal when rounded. We trained our supervised learning model to predict the risk score and compared the model predictions with the scores assigned by a highly trained medical professional. We found that using an ordinal dependent variable results in a model that is more robust and accurate than multinomial models using categorical variables (unordered risk classes).

The ML algorithm design was further specified to peanut. Focus on peanut was necessary as it was one of the most common food anaphylactic proteins in the study population. The

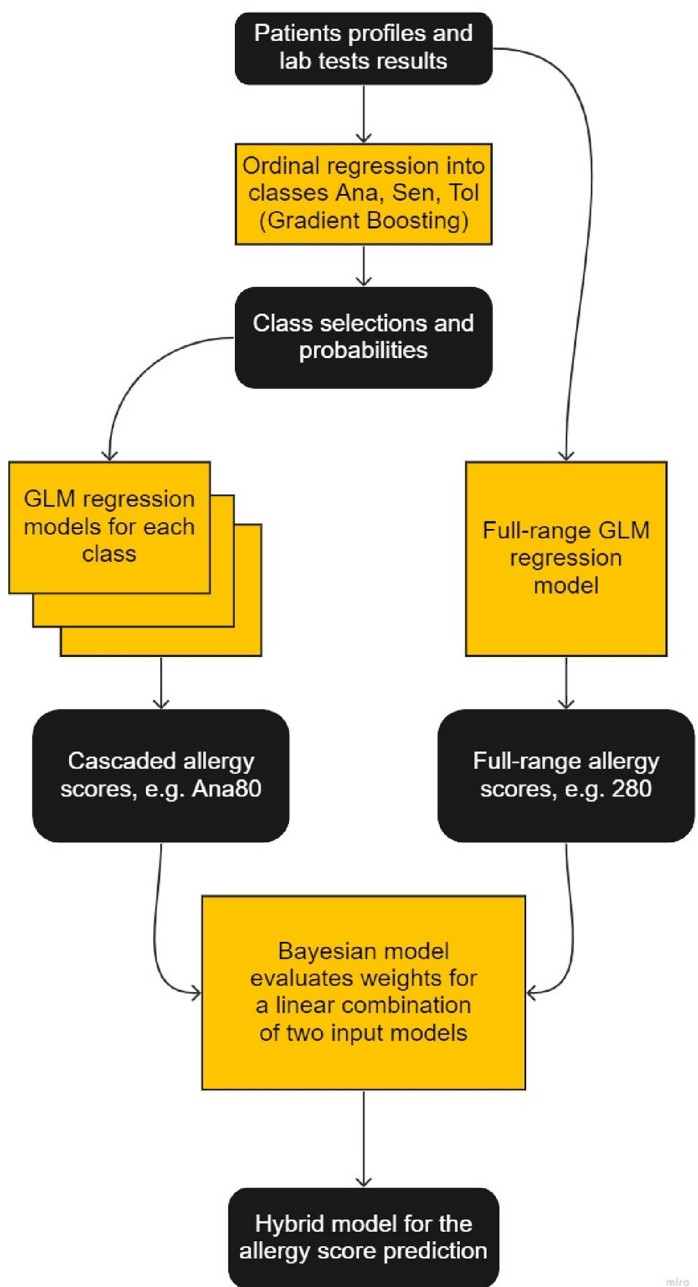

**Fig 4. The design of the proposed ML algorithm for prediction of the peanut allergy score.**

specific analysis of peanut anaphylaxis within this one endotype model was estimated to predict the severity of possible anaphylactic reaction to peanut with an overall recall of 95.2% on a dataset of 530 juvenile patients with various food allergies. The validation of the ML algorithm was characterized with ROC curves. Calculation of ROC curves for 340 patients with total IgE < 1000 kU/L and 190 patients with total IgE > 1000 kU/L yielded over 99% AUC (area under curve) results within peanut allergy prediction. The results demonstrate that the proposed ML algorithm could be used as a diagnostic support tool for a physician to evaluate the

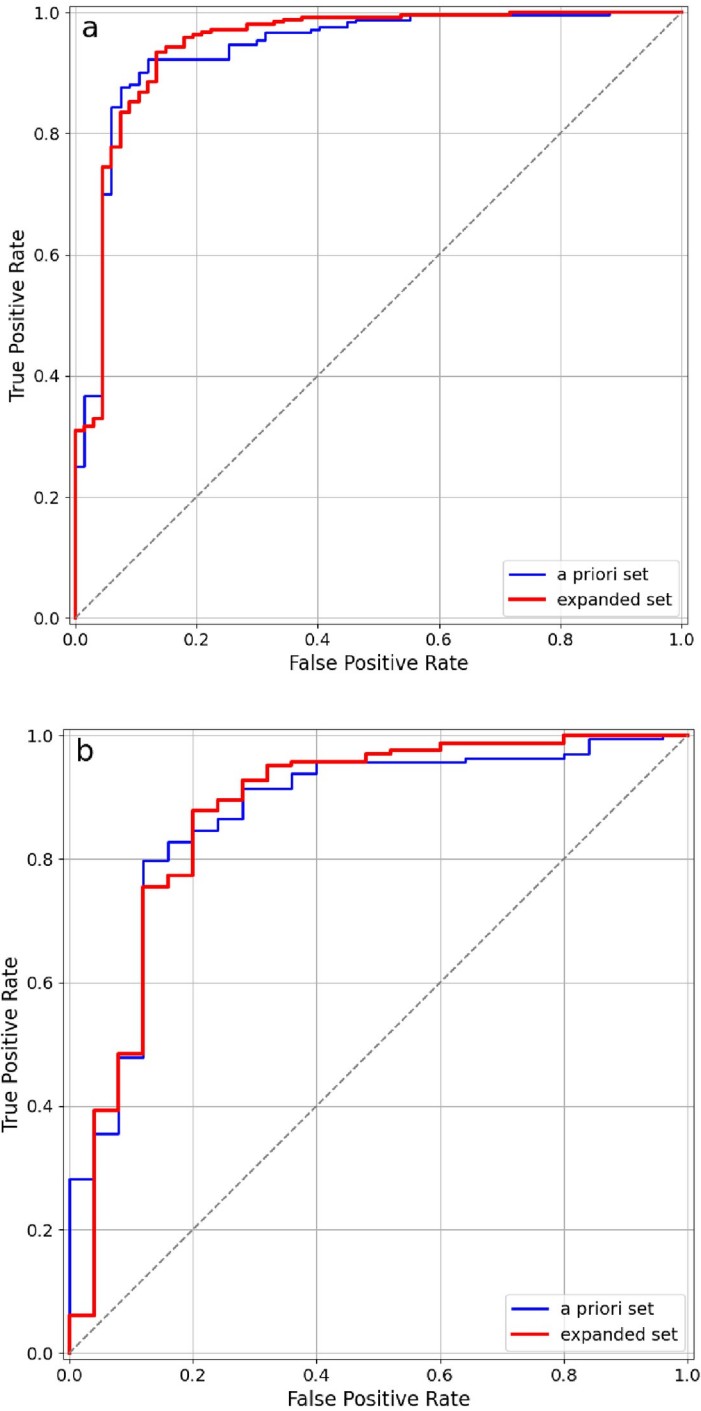

**Fig 5. ROC curves for the proposed ML algorithm's prediction that the patient is Anaphylactic (Ana) vs Sensitive (Sen).** (a) patients with total IgE < 1000 kU/L, AUC 0.994; (b) patients with total IgE > 1000 kU/L, AUC 0.997 (AUCs are given for the expanded feature sets).

expected severity of patient's allergic response to peanut. Moreover, it supports the accuracy of peanut anaphylaxis assessment in targeted treatment in the Tolerance Induction Program.

Machine learning algorithm design established from comprehensive molecular allergy data produces high accuracy and recall in anaphylaxis risk assessment. Further design of additional food protein anaphylaxis algorithms is needed to improve the precision and efficiency of clinical food allergy assessment and immunotherapy treatment.

## Author Contributions

**Conceptualization:** Inderpal S. Randhawa.

**Data curation:** Inderpal S. Randhawa, Kirill Groshenkov.

**Formal analysis:** Inderpal S. Randhawa.

**Software:** Kirill Groshenkov.

**Supervision:** Inderpal S. Randhawa.

**Visualization:** Grigori Sigalov.

**Writing – original draft:** Inderpal S. Randhawa, Grigori Sigalov.

**Writing – review & editing:** Inderpal S. Randhawa.

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
