## [Decision Letter · Decision Letter 0]

24 Oct 2022

PONE-D-22-23228Food Anaphylaxis Diagnostic Marker Compilation in Machine Learning Design and ValidationPLOS ONE

Dear Dr. Randhawa,

Thank you for submitting your manuscript to PLOS ONE. After careful consideration, we feel that it has merit but does not fully meet PLOS ONE’s publication criteria as it currently stands. Therefore, we invite you to submit a revised version of the manuscript that addresses the points raised during the review process.

We look forward to receiving your revised manuscript.

Kind regards,

Mohammad Asghari Jafarabadi

Academic Editor

PLOS ONE

Journal Requirements:

2. In the Methods section of your revised manuscript, please include the full name of the institutional review board or ethics committee that approved the protocol, the approval or permit number that was issued, and the date that approval was granted.

Please note that PLOS ONE has specific guidelines on code sharing for submissions in which author-generated code underpins the findings in the manuscript. In these cases, all author-generated code must be made available without restrictions upon publication of the work. Please review our guidelines at https://journals.plos.org/plosone/s/materials-and-software-sharing#loc-sharing-code and ensure that your code is shared in a way that follows best practice and facilitates reproducibility and reuse.

5. Please ensure that you include a title page within your main document. You should list all authors and all affiliations as per our author instructions and clearly indicate the corresponding author.

6. Please amend your manuscript to include your abstract after the title page.

8. Please include your tables as part of your main manuscript and remove the individual files. Please note that supplementary tables (should remain/ be uploaded) as separate "supporting information" files

Reviewers' comments:

Reviewer's Responses to Questions

**Comments to the Author**

1. Is the manuscript technically sound, and do the data support the conclusions?

Reviewer #1: Yes

Reviewer #2: Partly

2. Has the statistical analysis been performed appropriately and rigorously? 

Reviewer #1: Yes

Reviewer #2: No

3. Have the authors made all data underlying the findings in their manuscript fully available?

Reviewer #1: No

Reviewer #2: No

4. Is the manuscript presented in an intelligible fashion and written in standard English?

Reviewer #1: Yes

Reviewer #2: No

5. Review Comments to the Author

Reviewer #1: Great topic, very motivating and intriguing.

I have the following comments for authors to consider:

1) what's authors recommendation as to using ordinal regression or using pure classification and let the model decide? I personally think using ordinal regression make better sense, first because the three classes follow increase of severity, i.e. ordinal scale, second using ordinal regression should increase model prediction accuracy and model efficiency with reduced model complexity.

2) I think a lot of readers want to know, out of the 200 features which ones are the most important, e.g. top20, top10? and what's the AUC based on reduced features, such as top 20, top 10 features?

3) which software is the algorithm being built? will they be publicly available?

4) I think a realistic dataset should consist of patients with no peanut allergy and patients with different severity levels of peanut allergy. Can authors confirm their dataset takes this into consideration?

5) how ready is this approach in real world applications? I want to hear about authors' vision on this.

6) If the word limit still allows, I suggest authors expand the manuscript to have more details as appropriate and can be presented more clearly e.g. using bullet points to delineate the major steps

7) I think authors should consider finding a completely independent dataset as a testing dataset, to further evaluate the robustness of the trained model.

Reviewer #2: Question 1

In general, the main elements of the design of the automatic learning model are well listed (data, processing, prediction result). The scientific language can be improved (I noted this in the reviewed manuscript).

The data included in the model are quite clearly specified.

The data processing mode is relatively well specified.

The final result should be, according to the Methods and Results chapter, the prediction of the allergen score using 247 individual allergy analyses per patient. The prediction can be done through various regressions. The authors should specify the form of these predictions. In this way, we could understand to what extent each independent parameter considered (...) influences the allergen score. Therefore, the authors are asked to specify how the allergen score is calculated as a depending on the independent parameters introduced. The authors clearly show how the independent parameters are quantified, but not the regressions for the calculation of the dependent parameter: the allergen score.

The result of the analysis can also be materialized in a support decision, which will suggest to the doctor the probable decision. However, it must be specified how this decision is made by the car. Both types of results must be validated. To describe an algorithm means to give its results and, in order to be credible, the results of its validation must be given.

Question 2

In the Chapter Data and features the independent data used are listed:

• Patient's Personal Information, Including Gender, Age, and Weight;

• The Results of Up to 247 Blood and Skin Lab Tests Performed;

and the dependent data:

• Peanut Allergy Severity Class and Score as determined by a Qualified Medical Professional.

It is not a clear if they are accessible to the reader in gross form, in order to verify the graphic and functional representations.

I could not find in the manuscript received tables 1 and 2, which makes the text less clear, but in the first impossible to publish without these tables.

For example, the chapter Data and features clearly show what are the independent parameters. A problem that would clearly make the author clarified is if the decision tool useful to the allergist doctor consists of a relationship of the type:

- Peanut Allergy Severity Class and Score (gender, age, weight, blood lab tests, skin lab tests) = a*gender+b*Age+c*Weight+D*Blood Lab Tests+F*Skin Lab Tests

or

- in a decision scheme that, following the introduction of data required in the computer, decides with yes or no whether the patient presents a risk of allergy or not, possibly a value that indicates the intensity of the risk on a numerical scale.

Question 3

In the data chapter, the author specifies two types of data:

d1) All the molecular data that have been constantly acquired from each patient registered in the 2007 program: extended profiles of the component diagnosis, testing of the skin and in all primary plants and animal proteins, regardless of the patient's clinical history, the clinical history of the patient;

D2) Data includes markers of the immune system constantly acquired from each patient registered in the 2007 program: B diversity, cytokine production, basophilic indices, number of eosinophils and number of lymphocytes;

At the end of the given chapter, the author writes about the "final data set", which "includes the resemblance to allergens of plant and animal species that use data adopted from public fields, such as allergen nomenclature." That is understood by the "resemblance" with allergens ...? Then the author gives two databases at two internet addresses.

Unclear is the following: the data from the first two sets of data, are archived in a public database. Specifically, are the data accessed by the ML algorithm archived in the gross form in the public databases? Otherwise, how can this data be accessed by the reading public to verify the statements in the chapter of discussion and conclusions?

More precisely (now I mean the data and features chapter), where the readers can access the independent data listed in the Data and Features chapter:

• Patient's Personal Information, Including Gender, Age, and Weight;

• The Results of Up to 247 Blood and Skin Lab Tests Performed;

• Peanut Allergy Severity Class and Score as determined by A Qualified Medical Professional?

Question 4

In general, the scientific language can be adjusted, so that it is accessible to the non -specialists, at least partially. I noted in the manuscript reviewed some of these problems.

There are some abbreviations that are not explained. Maybe for specialists, it is not necessary, but for the general public (I refer to doctors with specializations of a different type) I think it is good to be explained. I noted in the manuscript reviewed some of these problems.

It is recommended to control the English language with the text publisher and another specialized publisher (for example Grammarly, free or licensed form).

An important problem to make the published article is that of the list of references and their summons in the text. I have not been able to identify in the text the quotation only for reference 9, citation I think, is inadequate as the form. The cities of the other sources must be highlighted, otherwise, they do not have their place on the list. The style of listing and citing the editor's references must be checked (it seems "Vancouver", in this case).

Question 5

The manuscript proposed for publication presents the results of interesting and useful research, with immediate applications, which must be validated over time, as the doctors will use the application.

As the main area of belonging, the research of a very wide category of research and applications, namely: Machine Learning in the Medical Field (Literature is very rich in this scientific field and is of more and more advanced quantification director of medical diagnosis). Originality comes from limiting the problem of allergies and more specific, research is done on patients suffering from allergies to peanuts proteins.

It is proposed to be published after a major review in which the following problems must be solved: 1) inclusion in the text of tables 1 and 2; 2) formatting the list of references to the editor's requirements, including the cities, so that the cities of the works included in the reference list are obvious. The unexpected works in the text are usually not admitted in the list of references; 3) clarification of the independent and dependent variables considered and the shape or shape of the ML results, usable in the practice of medical diagnosis as well as the clarifications required on the databases used for designing the algorithm and training the model; 4) refinement of scientific language; 5) a control of English (UK); 6) Clear explanation of the validation of the proposed algorithm (was done on the data with which the model was built, were other data used?) 7) In conclusions, a forecast should be exposed on the period of time necessary for acquiring the confidence among allergic doctors. Possible continuation directions may include expanding the database and applying the algorithm in other medical fields.

Lucrarea este valoroasă și se propune să fie publicată după o revizuire care răspunde la cele șapte probleme enumerate mai sus.

6. PLOS authors have the option to publish the peer review history of their article (what does this mean?). If published, this will include your full peer review and any attached files.

Reviewer #1: No

Reviewer #2: **Yes: **Petru Cardei

---

## [Author Response · Author response to Decision Letter 0]

7 Dec 2022

All responses to reviewers are included in the uploaded document.

---

## [Decision Letter · Decision Letter 1]

18 Jan 2023

PONE-D-22-23228R1Food Anaphylaxis Diagnostic Marker Compilation in Machine Learning Design and ValidationPLOS ONE

Dear Dr. Randhawa,

Thank you for submitting your manuscript to PLOS ONE. After careful consideration, we feel that it has merit but does not fully meet PLOS ONE’s publication criteria as it currently stands. Therefore, we invite you to submit a revised version of the manuscript that addresses the points raised during the review process.

We look forward to receiving your revised manuscript.

Kind regards,

Mohammad Asghari Jafarabadi

Academic Editor

PLOS ONE

Journal Requirements:

Reviewers' comments:

Reviewer's Responses to Questions

**Comments to the Author**

1. If the authors have adequately addressed your comments raised in a previous round of review and you feel that this manuscript is now acceptable for publication, you may indicate that here to bypass the “Comments to the Author” section, enter your conflict of interest statement in the “Confidential to Editor” section, and submit your "Accept" recommendation.

Reviewer #1: All comments have been addressed

Reviewer #2: All comments have been addressed

2. Is the manuscript technically sound, and do the data support the conclusions?

Reviewer #1: Yes

Reviewer #2: Partly

3. Has the statistical analysis been performed appropriately and rigorously? 

Reviewer #1: Yes

Reviewer #2: Yes

4. Have the authors made all data underlying the findings in their manuscript fully available?

Reviewer #1: Yes

Reviewer #2: Yes

5. Is the manuscript presented in an intelligible fashion and written in standard English?

Reviewer #1: Yes

Reviewer #2: (No Response)

6. Review Comments to the Author

Reviewer #1: Thanks for addressing the comments. I think the authors should mention in discussion section that their algorithm works for ordinal variable, but might not work well for categorical/multinomial variables according to response to my question 1, so that people from other background who want to try this tool are fully aware of the limitation.

Reviewer #2: 1) Of the 9 works on the reference list, only the work [9] is cited in the text of the manuscript. In general, publishers demand that all the works from references be quoted in the text.

2) Even if the authors of the article state that the article is because they know all the abbreviations, I believe that the following abbreviations should be explained:

AUC, RAST, GLM.

Although the authors show that the article is for specialists who know the terms, you should not remove readers who read the article to try to use the techniques exposed in other scientific fields. However, the article contains specialized terms in biology and computer science. I suppose there are few specialists well familiar with both areas, so an explanation of the main specialized (specific) terms is timely.

7. PLOS authors have the option to publish the peer review history of their article (what does this mean?). If published, this will include your full peer review and any attached files.

Reviewer #1: No

Reviewer #2: **Yes: **Petru Cardei

---

## [Author Response · Author response to Decision Letter 1]

13 Feb 2023

The response is noted in the rebuttal letter.

---

## [Decision Letter · Decision Letter 2]

3 Mar 2023

Food Anaphylaxis Diagnostic Marker Compilation in Machine Learning Design and Validation

PONE-D-22-23228R2

Dear Dr. Randhawa,

We’re pleased to inform you that your manuscript has been judged scientifically suitable for publication and will be formally accepted for publication once it meets all outstanding technical requirements.

Kind regards,

Mohammad Asghari Jafarabadi

Academic Editor

PLOS ONE

Reviewers' comments:

Reviewer's Responses to Questions

**Comments to the Author**

1. If the authors have adequately addressed your comments raised in a previous round of review and you feel that this manuscript is now acceptable for publication, you may indicate that here to bypass the “Comments to the Author” section, enter your conflict of interest statement in the “Confidential to Editor” section, and submit your "Accept" recommendation.

Reviewer #1: All comments have been addressed

Reviewer #2: All comments have been addressed

2. Is the manuscript technically sound, and do the data support the conclusions?

Reviewer #1: Yes

Reviewer #2: Yes

3. Has the statistical analysis been performed appropriately and rigorously? 

Reviewer #1: Yes

Reviewer #2: Yes

4. Have the authors made all data underlying the findings in their manuscript fully available?

Reviewer #1: Yes

Reviewer #2: Yes

5. Is the manuscript presented in an intelligible fashion and written in standard English?

Reviewer #1: Yes

Reviewer #2: Yes

6. Review Comments to the Author

Reviewer #1: Thanks for addressing the comments. I think it will have a lot of good impact and keep up the good work.

Reviewer #2: Dear Authors

Referring to:

PONE-D-22-23228R2

In the next table, you have all my observations regarding the article. Essentially it can be published, but please consider the opportunity of the following suggestions.

Suggestions and observation to revision 2 for the paper:

Obs1) We ask the authors to specify if in fig. 1, 2 and 3, on the vertical axis, read the frequency or frequency logarithm. If the frequency logarithm is, please specify the base of the logarithm.

Obs2) It is recommended to control the English language for the entire manuscript because there are still control programs that claim errors and simplest expressions:

1) https://quillbot.com/grammar-check

2) https://www.grammarly.com/gramar-check

Obs3) Plagiaristic check with

https://smallseotools.com/plagiarism-checker/

show that:

1- Introduction 13% - plagiarism. Some sources are specified in references, others are not in the reference list. The sources that are not in references can be included, which would also increase the list of references.

2- Material and Methods, Data 10% plagiarism. A source not included in the references. Can be included.

3- A Machine Learning algorithm for allergen score prediction , 3% plagiarism. . A source not included in the references. Can be included.

Petru Cardei

7. PLOS authors have the option to publish the peer review history of their article (what does this mean?). If published, this will include your full peer review and any attached files.

Reviewer #1: **Yes: **Shuai Wang

Reviewer #2: **Yes: **Petru Cardei

---

## [Editor Report · Acceptance letter]

14 Mar 2023

PONE-D-22-23228R2 

Food anaphylaxis diagnostic marker compilation in machine learning design and validation 

Dear Dr. Randhawa:

I'm pleased to inform you that your manuscript has been deemed suitable for publication in PLOS ONE. Congratulations! Your manuscript is now with our production department. 

Kind regards, 

on behalf of

Professor Mohammad Asghari Jafarabadi 

Academic Editor

PLOS ONE